# The Perspective of Vitamin D on suPAR-Related AKI in COVID-19

**DOI:** 10.3390/ijms231810725

**Published:** 2022-09-14

**Authors:** Tzu-Hsien Liao, Hsien-Chang Wu, Min-Tser Liao, Wan-Chung Hu, Kuo-Wang Tsai, Ching-Chieh Lin, Kuo-Cheng Lu

**Affiliations:** 1Department of Chinese Medicine, Taipei Tzu Chi Hospital, Buddhist Tzu Chi Medical Foundation, New Taipei City 231, Taiwan; 2School of Post-Baccalaureate Chinese Medicine, Tzu Chi University, Hualien 970, Taiwan; 3Department of Pediatrics, Taoyuan Armed Forces General Hospital Hsinchu Branch, Hsinchu City 300, Taiwan; 4Department of Pediatrics, Tri-Service General Hospital, National Defense Medical Center, Taipei 114, Taiwan; 5Department of Clinical Pathology and Medical Research, Taipei Tzu Chi Hospital, Buddhist Tzu Chi Medical Foundation, New Taipei City 231, Taiwan; 6Department of Research, Taipei Tzu Chi Hospital, Buddhist Tzu Chi Medical Foundation, New Taipei City 231, Taiwan; 7Department of Chest Medicine, Taoyuan Armed Forces General Hospital Hsinchu Branch, Hsinchu City 300, Taiwan; 8Division of Nephrology, Department of Medicine, Taipei Tzu Chi Hospital, Buddhist Tzu Chi Medical Foundation, New Taipei City 231, Taiwan; 9Division of Nephrology, Department of Medicine, Fu-Jen Catholic University Hospital, School of Medicine, Fu-Jen Catholic University, New Taipei City 242, Taiwan

**Keywords:** AKI, COVID-19, suPAR, uPAR, vitamin D

## Abstract

The coronavirus disease 2019 (COVID-19) pandemic has claimed the lives of millions of people around the world. Severe vitamin D deficiency can increase the risk of death in people with COVID-19. There is growing evidence that acute kidney injury (AKI) is common in COVID-19 patients and is associated with poorer clinical outcomes. The kidney effects of SARS-CoV-2 are directly mediated by angiotensin 2-converting enzyme (ACE2) receptors. AKI is also caused by indirect causes such as the hypercoagulable state and microvascular thrombosis. The increased release of soluble urokinase-type plasminogen activator receptor (suPAR) from immature myeloid cells reduces plasminogen activation by the competitive inhibition of urokinase-type plasminogen activator, which results in low plasmin levels and a fibrinolytic state in COVID-19. Frequent hypercoagulability in critically ill patients with COVID-19 may exacerbate the severity of thrombosis. Versican expression in proximal tubular cells leads to the proliferation of interstitial fibroblasts through the C3a and suPAR pathways. Vitamin D attenuates the local expression of podocyte uPAR and decreases elevated circulating suPAR levels caused by systemic inflammation. This decrease preserves the function and structure of the glomerular barrier, thereby maintaining renal function. The attenuated hyperinflammatory state reduces complement activation, resulting in lower serum C3a levels. Vitamin D can also protect against COVID-19 by modulating innate and adaptive immunity, increasing ACE2 expression, and inhibiting the renin–angiotensin–aldosterone system. We hypothesized that by reducing suPAR levels, appropriate vitamin D supplementation could prevent the progression and reduce the severity of AKI in COVID-19 patients, although the data available require further elucidation.

## 1. Introduction

Approximately 75% of all patients with coronavirus disease 2019 (COVID-19) have acute kidney injury (AKI) and/or abnormal urine dipstick findings [1]. The analysis of some, although not all, urine and postmortem kidney tissue revealed the presence of severe acute respiratory syndrome coronavirus 2 (SARS-CoV-2), confirming the kidney is one of the main targets of COVID-19 [2]. Epidemiological studies in Europe and China have reported AKI in approximately half of the patients with fulminant sepsis [3,4]. Another clinical study identifying resuscitation strategies for severe sepsis also reported that AKI occurs in approximately half of all emergency department patients [5]. Furthermore, almost one-third of patients with community-acquired pneumonia develop AKI even without evidence of severe sepsis [6]. Before the pandemic, a clue showed the incidence of COVID-19-associated AKI exceeds that of sepsis-associated AKI. However, the incidence of AKI was not significantly different from other forms of sepsis over time.

The uPA (urokinase plasminogen activator) and uPA receptor (uPAR or CD87) form a multimolecular complex at the cell surface which can cause fibrinolysis and perform immune functions [7]. The uPAR may be detached from the cellular membrane glycosylphosphatidylinositol (GPI) anchorage, resulting in the release of the soluble form of uPAR (suPAR) [8]. The uPA localized in podocytes/tubular cells can bind to the uPAR domain, then the uPAR can interact with other molecules such as integrins and vitronectin. Damaged endothelial cells produce and secrete plasminogen activator-inhibitor 1 (PAI-1). The binding of PAI-1 to the uPA-uPAR complex can inhibit plasminogen activation [9]. In COVID-19, excess suPAR may bind to uPA, thus preventing uPA from activating plasminogen [10]. By binding to uPAR, uPA catalyzes the conversion of plasminogen to plasmin, which contributes to the breakdown of the extracellular matrix (ECM) and promotes cell motility [11]. Once the uPA binds to the uPAR, a structure configuration transition occurs and it can play a role in linking with other co-receptors such as integrins and vitronectin. The uPAR acts on αvβ3 integrins and can promote cell motility in kidney podocytes [12]. The binding of the vitronectin with uPAR and β3 integrins is necessary for the activation of catabolic proteases such as MMP-3 and MMP-13 [13].

The suPAR has a potential prognostic value in critically ill patients [14]. In clinical laboratory medicine, it has the potential to predict critical illness and renal failure in intensive care unit (ICU) patients [15]. Low levels of suPAR have been reported as a positive predictor of critical care survival in seriously ill patients [16]. Afro-Caribbean patients with COVID-19 and increased suPAR levels exhibit a high risk of AKI [17]. Because elevated levels of suPAR can damage kidney cells directly, they can also increase the incidence of hypofibrinolysis and thrombogenesis, which can contribute to the severity of AKI. In COVID-19 inpatients, circulating suPAR levels on admission can predict the development of AKI and the need for renal replacement therapy [18]. 

Vitamin D deficiency is associated with severe COVID-19 in adult patients [19]. In children, a potential association between a profound vitamin D deficiency and the severity of COVID-19 multisystem inflammatory syndrome (C-MIS) has been reported [20]. Vitamin D levels can be used to predict the risk for severe C-MIS, and correcting abnormal levels in severe cases may help reduce the severity of C-MIS [21]. Vitamin D can attenuate the probability of breast tumor metastasis by inhibiting uPAR expression [22]. Vitamin D can also prevent podocyte injury and proteinuria development by inhibiting podocyte uPAR expression [23]. In mice with chronic kidney disease (CKD) and septicemia, treatment with vitamin D was found to inhibit podocyte uPAR expression and provide an obvious anti-proteinuric effect [24]. The renal tissue study of vitamin D receptor knockout diabetic mice showed that the glomerular basement membrane was significantly thickened and the podocytes were significantly reduced, resulting in more severe glomerulosclerosis and obvious proteinuria [25]. We hypothesized that the potential beneficial effects of vitamin D can be directly mitigated through uPA/uPAR-related integrin signaling and indirect anti-inflammatory response. However, the molecular mechanism by which vitamin D inhibits podocyte uPAR in COVID-19-related AKI remains to be fully elucidated. Therefore, we conducted this narrative review to explore the potential role of vitamin D in preventing AKI in COVID-19 patients. We specifically focused on the uPA/suAPR pathway and its possible relationship with vitamin D in patients with COVID-19-related AKI.

## 2. COVID-19-Associated AKI

### 2.1. Clinical Manifestations

Nearly 5% of patients with COVID-19 have serious associated conditions such as septicemia or organ failure. Proteinuria, hematuria, and AKI are the main features of kidney injury [26]. Kidney damage caused by COVID-19 manifests as increased serum levels of creatinine, proteinuria, hematuria, and AKI, and some patients require renal replacement therapy (RRT) [1,27]. Based on KDIGO’s (Kidney Disease: Improving Global Outcomes) AKI criteria, approximately half of such patients are in stage 3 [28]. In a cohort study, a high incidence of proteinuria and hematuria was reported and linked to mortality in patients with critical COVID-19 [29]. The qualitative and quantitative assessments of renal impairment are performed by determining decreased renal parenchymal attenuation on non-enhanced computed tomography, which shows a weak linear and inverse correlation with serum creatinine levels in COVID-19 patients [30].

Previous retrospective studies have shown that AKI is common in hospitalized patients due to COVID-19 and is associated with elevated mortality. The kidney function at discharge is restored in only 30% of patients with AKI [31]. An elevated incidence of AKI is found in patients with other comorbidities such as high blood pressure, diabetes, and pre-existing kidney disease [32]. Kidney function was not restored to baseline in one-third of patients with AKI and COVID-19. Considering the overall severity of AKI and the high prevalence of severe tubular lesions, underlying vascular micro-thrombus, and protein in COVID-19 patients, lower renal recovery rates are expected [33]. Another retrospective observational study evaluating clinical outcomes in adults with AKI with COVID-19 suggested that vital signs and laboratory data at admission may be useful for the risk stratification of severe AKI. The best-known risk factors associated with COVID-19-mediated AKI are obesity, diabetes, hypertension, aging, and pre-existing CKD [34].

### 2.2. Histopathological Characteristics and Pathophysiology

Initial histopathological analyses of 26 autopsy samples obtained from the kidneys of patients with COVID-19 provide direct evidence of SARS-CoV-2 invasion of the kidney tissue [33]. SARS-CoV-2 can cause kidney damage in a variety of ways, and it can directly infect renal tubular cells and podocytes through the ACE2 receptor, resulting in increased mitochondrial/endoplasmic reticulum (ER)-related oxidative stress, acute tubular necrosis, Bowman’s capsule leak, and collapsing glomerulopathy. Renal damage in patients with preexisting renal disease and increased expressions of ACE2 and CD147 in the renal tissue may be aggravated [35]. The histological analysis of renal biopsy samples from patients with COVID-19 showed tubular epithelium with reactive nuclei, significantly simplified cytoplasm, and obvious denudation of brush borders. Glomeruli showed a sagging of the tuft and hypertrophy and hyperplasia of mesangial cells (MCs) and epithelial cells. The ultrastructural study revealed extensive foot process effacement and tubuloreticular inclusions in a glomerular endothelial cell [36,37].

The detailed pathophysiology of COVID-19-related AKI is unclear [38]. Dysregulated immune reactions induced by SARS-CoV-2 may be a cause of AKI. In addition, endothelial dysfunction, prominent hypercoagulability, and microbial infection-related sepsis are other important potential causes of AKI. Furthermore, hypoxemia in the kidneys may lead to ischemic damage [39]. Potentially modifiable factors such as complement activation, hypercoagulability/hypofibrinolysis, and hyperinflammatory reaction should be explored in detail [34,40]. Whether targeting these pathways using vitamin D, anticoagulants, or cytokine inhibitors will reduce the severity of AKI is unclear. Moreover, the detailed mechanisms of antiviral therapy in the manipulation of COVID-19-related AKI remain unclear [41,42].

Hypercoagulability and microvascular thrombus formation are common mechanisms for acute respiratory distress syndrome and AKI. Immunofluorescence tests have shown the SARS-CoV-2 protein in different renal cells. However, D-dimer concentration has not been directly linked with AKI [38,43]. AKI in COVID-19-related sepsis has a multifactorial etiology, and suPAR is a crucial potential mediator [44,45]. As AKI is associated with local intrarenal and systemic inflammation and intravascular hypercoagulability; therefore, further exploration of the underlying molecular responses may help in the search for effective treatments to reduce the severity of AKI.

#### 2.2.1. Direct Effects of SARS-CoV-2 on AKI

AKI observed in the context of COVID-19 is associated with a direct viral invasion of renal cells. The invasion of renal cells by SARS-CoV-2 is mediated by ACE2 receptors, resulting in acute tubular necrosis, filtration barrier disruption, and micro thrombosis with collapsing glomerulopathy [39,46]. The entry of SARS-CoV-2 into cells requires the interaction of ACE2, TMPRSS2 (transmembrane protease serine 2), ADAM17 (a disintegrin, metalloproteinase 17), and possibly cell membrane CD147 [47]. TMPRSS2 expression is lower in proximal tubules; therefore, its role in regulating the priming process is unclear [48]. Distal tubules express sufficient TMPRSS2 to facilitate the entry of SARS-CoV-2 into host cells [49]. With the help of TMPRSS2, the SARS-CoV-2 spike protein interacts with the ACE2 in the cell membrane, thereby promoting viral internalization. ADAM17 can eliminate circulating SARS-CoV-2 by splitting the ACE2 in the membrane into a soluble form which can capture SARS-CoV-2 particles. However, the expression of CD147 is primarily distributed on the basolateral side of the proximal tubes. Enhanced CD147 expression can further promote SARS-CoV-2 entry into renal tubular cells with pre-existing lesions [50].

Conventional renin–angiotensin–aldosterone system (RAAS) activation is common in CKD patients, and localized reduction in ACE2 expression increases hypercoagulability and induces the formation of microthrombi [51]. In the kidney, glomerular-filtered SARS-CoV-2 can invade proximal tubular epithelial cells via luminal ACE2 receptors. Entrance via luminal or basolateral routes can exacerbate tubule-interstitial inflammation and decrease cell survival. Damaged glomerular podocytes increase CD147 expression, which further enhances SARS-CoV-2 invasion. After SARS-CoV-2 enters the host cell, the virus downregulates the expression of ACE2 in the host cell, thereby increasing the expression of angiotensin II (Ang II) [52]. In addition to increased blood pressure, Ang II through nuclear factor κB signaling will enhance the expression of several inflammatory cytokine genes [53]. Over-activation of the RAAS can further enhance the expression of CD147. Therefore, increasing SARS-CoV-2 entry and inducing podocyte damage is one of the key mechanisms for inducing AKI [35]. Previous reports have also suggested that genetic variation in the ACE receptor is another key factor. In addition, the genotype of apolipoprotein L1 variant (APOL1) is another genetic risk factor for collapsing glomerulopathy in COVID-19 patients [54,55]. 

#### 2.2.2. Indirect Effects of SARS-CoV-2 on AKI

Injured lung from COVID-19 releases damage-associated molecular patterns (DAMPs) and pathogen-associated molecular patterns (PAMPs) into circulation, which are then filtered in the glomerulus and can interact with pattern recognition receptors (PRRs) on tubular epithelial cells, resulting in renal tubular epithelial damage. DAMPs also elicit systemic inflammation that can damage the kidneys and other organs [56]. The release of chemokines can attract additional inflammatory cells to the site of inflammation, thereby exacerbating the cytokine storm, and can have indirect effects on several organs, especially on the kidneys [39]. Some COVID-19 patients experience the fulminant activation of macrophages following a cytokine storm. They often have high levels of ferritin in the plasma [57]. In addition, myocardial lesions secondary to pulmonary lesions can develop in critically ill patients [58]. Interactions between damaged lungs, heart, and kidneys can adversely affect clinical outcomes. 

In addition, potential nephrotoxic drugs are commonly used in critical patients with COVID-19 [59]. An increase in right heart ventricle pressure can result from left heart ventricle failure or high positive expiratory pressure on mechanical ventilation in patients with respiratory failure. Mitochondrial dysfunction induced by SARS-CoV-2 can damage kidney cells directly and indirectly by enhancing systemic oxidative stress and inflammation [60]. Other epigenetic alterations, such as those in miRNAs, influence the pathogenesis of COVID-19 [61]. Consequently, the molecular mechanisms of AKI in COVID-19 patients would be similar to those caused by other forms of sepsis.

## 3. Role of suPAR in COVID-19-Related AKI

Infected critically ill patients usually have higher systemic suPAR concentrations than uninfected patients [16,62]. A meta-analysis of biological markers in adults disclosed that suPAR can be used as a diagnostic and prognostic marker for microbial infection [63]. However, the role of suPAR in critically ill patients with AKI is unclear [64]. An observational study has also shown that elevated suPAR levels predict AKI and the need for dialysis. As a result, suPAR may play an important role in the pathophysiology of AKI in COVID-19 [18].

### 3.1. Function of the Urokinase Receptor System

The uPAR is a glycosylphosphatidylinositol (GPI)-anchored urokinase receptor. The uPA can bind to uPAR and integrin through a growth-factor-like domain and connecting peptide (Figure 1). Three different suPARs exist in the circulation (suPAR DI-III, DI, and DII-III). Among them, suPAR DI-III is considered the active form of suPAR due to its strong affinity for uPA binding [65]. A recent report found that a dimer form of the mouse isoform 2 uPAR can induce kidney disease [66]. The overexpression of this dimer can cause kidney disease in mice by activating glomerular Src kinase through β_3_ integrin signaling [66]. The activation of uPA and plasminogen-dependent latent transforming growth factor β1 (TGF-β) yields active TGF-β, a powerful driver of endothelial–mesenchymal transition in the kidney [67].

The uPA localizes in podocytes/tubular cells by binding to uPAR domain 1 (DI). The uPA domains are represented as the growth-factor-like domain (GFD), the kringle domain (KD), the interdomain linker or “connecting peptide” (CP, residues 132–158), and the serine protease domain (CD). The uPA interacts with the GPI-anchored uPAR through GFD and with integrin through CP, bridging the two receptors together [67]. The amino-terminal region of uPA can bind to the DI region of uPAR. The DII:DIII region of uPAR can interact with co-receptors such as the vitronectin and membrane integrin (αvβ3, αvβ5) families. The binding of plasminogen activator-inhibitor 1 (PAI-1) to the uPA–uPAR complex can not only inhibit the uPA activation of plasminogen but also promote the internalization of uPA–uPAR–PAI-1 complex and β1-integrin, which then recycles uPAR back to the cell surface [9]. Under physiological conditions, after binding to uPAR, the uPA can catalyze the conversion of the inactive plasminogen to active plasmin [11]. Several proteins expressed during fibrogenesis can be cleaved by plasmin. In addition, plasmin can also activate matrix metalloproteinases (MMPs) and activate or release growth factors such as hepatocyte growth factor. Once uPA binds to uPAR, the structure configuration will be changed, and the uPAR SRSRY (88–92) sequence can bind with other co-receptors (Figure 1). Chymotrypsin and cathepsin G can hydrolyze uPAR at the DI:DII junction region, resulting in fragmented DII:DIII GPI-anchored uPAR. This modified uPAR would then detach from the GPI anchor, resulting in the release of the soluble form of uPAR (suPAR). Since uPAR lacks an intracellular domain, it will interact with other transmembrane receptors such as formyl peptide receptors (FPRs) and integrins. FPRs can also be activated by the DII:DIII SRSRY fragment. Briefly, the activation of these co-receptors will elicit intracellular signaling which is related to the synthesis of pro-angiogenesis and pro-inflammatory mediators [8]. 

The arginine–glycine–aspartic acid (RGD)-containing vitronectin fragments can induce podocyte damage through β3 integrin signaling (Figure 1). In addition, protein kinase C-delta (PKCδ) is the rate-limiting factor at the signaling transduction from vitronectin, which can activate PKCδ. PKCδ can activate NF-κB and mitogen-activated protein kinase (MAPK) signaling. The activation of MAPKs (ERK1/2, JNK1/2, and p38) will inhibit anabolic signaling such as IGF-1 and BMP7 signaling, increasing inflammatory cytokine levels, and upregulating catabolic proteases such as MMP-3 and MMP-13 [13]. The uPAR can act on β3 integrins to enhance cell motility in kidney podocytes. However, the binding of vitronectin to uPAR and β3 integrins is required for the activation of this pathway. This uPAR–β3 integrin interaction activates Src kinases, which leads to the phosphorylation of the focal adhesion kinase (FAK)-associated adapter protein. The activation of Rac by Src or FAK can stimulate actin polymerization, eliciting cell motility. In mouse podocytes, the uPAR–αvβ3 integrin signaling can activate the Rho family GTPase Cdc42 in a vitronectin-dependent manner. However, uPAR signaling through αvβ3 integrin could be suppressed by uPA-mediated uPAR cleavage [12] (Figure 1).

Integrin is composed of non-covalently linked α and β subunits which can be categorized into laminin, collagen, and RGD-binding receptors [68]. In addition to anchoring cells to the ECM (extracellular matrix), integrins also signal bidirectionally through the plasma membrane [69].

In the kidney, major laminin-binding receptors include integrins α3β1, α6β1, and α6β4. Effective adhesion of podocytes to the glomerular basement membrane (GBM) requires integrin α3β1 [69]. There are two major collagen receptors in the kidney, integrin α1β1, and α2β1, which play a crucial pathogenic role in inflammatory forms of kidney damage. In MCs, the binding of integrin α2β1 to collagen activates intracellular signaling, resulting in the increased production of collagen and reactive oxygen species (ROS) [70]. The collagen receptors in the kidney, integrin α1β1, and α2β1, play an important pathogenic role in inflammatory forms of kidney damage. However, uPA/uPAR mainly acts on the αvβ3 integrin. The RGD-containing ligands such as fibronectin and vitronectin can bind to β1, αv, and αIIbβ3 integrins. The αv subunit of integrin is ubiquitously expressed in adult kidneys, and it can modulate glomerular extracellular matrix homeostasis. The activation of integrin αvβ3 by uPAR and suPAR in podocytes has been reported to result in podocyte dysfunction [23]. Since COVID-19-related AKI may have high serum suPAR levels, we speculate that these high suPARs can activate integrin signaling that may cause kidney injury. 

### 3.2. Putative Mechanism of suPAR-Related Kidney Injury in COVID-19-Related AKI

The suPAR is a circulating biomarker that can cause glomerular podocyte injury. The suPAR levels are increased in the plasma of COVID-19 patients, which provides mechanistic insights into how increased suPAR levels can lead to AKI. In this case, suPAR is produced mainly by immature Gr-1^lo^ myeloid cells, neutrophils, monocytes, and perhaps T lymphocytes. Through blood circulation, suPAR enters the glomerulus of the kidneys and binds and activates β3 integrin. High plasma levels of suPAR enhanced β3 integrin signaling activation, leading to podocyte dysfunction, effacement, and proteinuria. The activation of integrin αvβ3 by uPAR and suPAR in podocytes impairs the podocyte function [69]. The damaged podocytes repress vascular endothelial growth factor expression which leads to endothelial dysfunction. The damaged endothelial cells produce and secrete PAI-1, which binds to circulating uPA either in the capillaries or on the podocyte surface, which makes a complex with uPAR. The complex can bind to β1-integrin on the surface of the podocyte, then β1-integrin translocates into the cytoplasm through endocytosis. Then, the β1-integrin-lost podocytes will detach from the glomerular basement membrane [71]. 

In terms of microvascular thrombosis, elevated suPAR levels in COVID-19 and AKI are associated with increased extra mitochondrial enzyme oxidation and oxidative stress in renal cells. Increased suPAR release in the bloodstream reduces plasmin production by competitively inhibiting uPA function on the renal cell surface. This may promote hypofibrinolytic status in critically ill patients with COVID-19, which will synergistically exacerbate thrombosis severity [10]. Furthermore, increased C3a production in patients with severe COVID-19 induces the release of activated CD_16_^+^ cytotoxic T cells from bone marrow. The proportion of activated CD_16_^+^ T cells and plasma C3a levels are associated with lethal thromboembolic outcomes in COVID-19 (Figure 2).

The uPAR primary ligand is the uPA, which converts plasminogen into plasmin. The uPA activates plasminogen in solution or when associated with its cellular receptor uPAR in a fibrin-independent manner [72]. The uPAR is a multifunctional receptor that coordinates pericellular proteolysis related to cell membrane integrin signal transduction, affecting normal physiology and leading to abnormal pathological mechanisms [67] (Figure 1).

The uPA and uPAR are expressed by different cells in the hematopoietic lineage. Multiple observations linked the fibrinolytic system to innate and adaptive immunity [73]. In general, systemic inflammation resulting from various infections leads to the activation of the coagulation system through the over-production of thrombin, attenuation of anticoagulant mechanisms, and inhibition of fibrinolysis. The proinflammatory cytokines (IL-1β, TNFα) play a crucial role in both the coagulation and fibrinolysis pathways [74]. Severe or critical COVID-19 is associated with an increased secondary infection rate and could lead to a significantly worsened prognosis. Secondary infection risks increased after receiving invasive respiratory ventilation and intravascular devices. The most common infections were respiratory, and the most detected pathogens were gram-negative bacteria, followed by gram-positive bacteria, viruses, and fungi [75]. During bacterial infection, circulating levels of uPA and its inhibitor PAI-1 increase significantly. These increases are also triggered by releases of bacterial endotoxins and cytokines due to the activation of innate immunity [76]. The conversion of uPA-related plasminogen into plasmin promotes the release of proinflammatory cytokines and activates phagocytic monocyte/macrophage (PMM) zymogenes, thus amplifying acute inflammatory responses [77] (Figure 1). 

### 3.3. suPAR as a Biomarker of Infection in AKI

It is well known that circulating suPAR levels can represent the overall activity of the uPA/uPAR system that is related to innate immunity, proteolytic enzyme activity, and ECM remodeling capability [12,78]. The uPAR can be induced by leukocyte activation and differentiation in response to smoking and certain RNA viruses [79,80]. The suPAR has been identified as a functional link between the bone marrow (BM) and kidneys, and an increase in immature Gr-1^lo^ myeloid cells in the BM is a key factor in the subsequent development of glomerular dysfunction caused by increased suPAR in proteinuric mice [81]. Therefore, BM-derived immature myeloid cells serve as the primary source of suPAR and may contribute to renal pathologies [82].

#### 3.3.1. Effects of suPAR on Podocyte Injury

The suPAR is a risk factor in several human clinical conditions [14,83,84], such as AKI [85,86], and CKD [87]. In native and grafted kidneys, suPAR circulating has been shown to activate podocyte β integrin3, thus causing foot process erasure, proteinuria, and focal segmental glomerular glomerulopathy [88]. Hence, a high incidence of kidney disease is observed in patients with a high proinflammatory state with high levels of suPAR [89,90,91,92]. In podocytes, the induction of uPAR leads to foot process disappearance and the loss of urinary protein through the activation of αvβ3 integrin signaling [23] (Figure 2). High levels of circulating suPAR strongly predict renal function progression; further, chronic exposure directly activates podocyte αvβ3 integrin, resulting in proteinuria and ultimately, chronic renal dysfunction [66,81]. A previous study on transgenic mice with an over-expression of suPAR demonstrated the development of proteinuria and severe AKI after contrast injection [85].

Circulating suPAR affects podocyte configuration and viability, resulting in foot process effacement and cell apoptosis [88,93]. The structure–function association of suPAR with αvβ3 integrin can be regulated by sphingomyelinase-like phosphodiesterase 3b (SMPDL3b). SMPDL3b may modify the glomerular filtration barrier by affecting podocyte function due to the movement of a migratory phenotype to an apoptotic phenotype [94,95] (Figure 2). The lipid-modulating enzyme SMPDL3b is also a known off-target therapeutic of the anti-CD20 antibody rituximab [96]. 

A study of lipopolysaccharide (LPS)-mediated proteinuria in mice showed that uPAR is required for the activation of αvβ3 integrin signaling, enhancing the motility and activation of the small GTPases Cdc42 and Rac1 in podocytes (Figure 2). The blockade of αvβ3 integrin signaling will reduce podocyte motility and proteinuria in mice [23]. In addition, APOL1 gene variants G1 and G2 exhibit high affinity for activated αvβ3 integrin, which would synergize with suPAR to further activate αvβ3 integrin signaling on podocytes, thereby promoting the development of CKD [97].

#### 3.3.2. Effects of suPAR on Tubular Cell Injury

A study in mice with high expression of suPAR showed that they are vulnerable to contrast-mediated kidney injury. Further, proximal tubular cell cultures showed altered mitochondrial respiration with increased energy demand and the production of oxygen free radicals [85]. Elevated suPAR expression may induce tubulointerstitial fibrosis in an integrin-dependent manner. Versican is a large ECM proteoglycan found in different human tissues. In patients with focal segmental glomerular sclerosis (FSGS), renal tissue showed high levels of versican. Rienstra et al. showed that renal tubular-cell-derived versican can induce fibroblast proliferation and collagen production by activating the CD44/Smad3 pathway in renal interstitial fibroblasts [98]. In patients with COVID-19, an increased tubular bond between urinary C3a and suPAR enhances tubular versican expression and subsequently promotes renal interstitial fibrosis [99] (Figure 3).

#### 3.3.3. The αvβ3-Integrin-Dependent Activation and Mesangial Fibrosis

MCs are stromal cells and are important for glomerular homeostasis and reaction to injury [100]. The plasma-derived fibrinogen is deposited in the mesangium of certain renal lesions. The adhesion of MCs to fibrinogen is mediated by αvβ3 integrin [101]. The histological analysis of samples obtained from a kidney biopsy performed in a patient with COVID-19 showed mesangial hyperplasia with fibrocystic crescents in some glomeruli. Occasionally, various degrees of secondary sclerotic ischemic changes are seen in the mesangial stroma associated with GBM thickening and podocyte detachment [102]. This sclerotic change is associated with TGF-β1 signaling. In cultured renal MCs, TGF-β1 stimulates collagen expression through the αvβ3-integrin-dependent activation of FAK and subsequent ERK/MAPK activity. This causes the phosphorylation of the Smad3 link region, which leads to collagen production and mesangial fibrosis [103]. High PAI-1 expression in MCs activates the MAPK signaling pathway and stimulates TGF-β1 expression, leading to accumulation in the mesangial matrix [104]. In rats with diabetes mellitus, uPA can improve the diabetic mesangial condition by binding to uPAR for capturing PAI-1 and accelerating its degradation, thereby attenuating the expression of MCs and their matrix [105].

#### 3.3.4. Synergistic Effects of suPAR and C3a on Kidney Injury

In the kidney, human C3a and its receptor C3aR are mainly expressed in the proximal renal tubular cells [106,107]. In renal diseases, the expression of C3aRa and levels of C3a in plasma and urine are increased, and both are associated with the progression of renal disease. This reveals that the C3a/C3aR route facilitates the progression of glomerular and tubule-interstitial pathologies [108]. Patients infected with SARS-CoV-2 are at increased risk of dying due to the hyperactivity and consumption of C3 [109]. Dysregulated pro-inflammatory response may activate the complement system, which is associated with increased mortality and thromboembolic complications [110]. The increased production of C3a in severe COVID-19 can activate cytotoxic CD_16_^+^ T lymphocytes. Both CD_16_^+^ activated T-cell proportions and C3a plasma concentrations are associated with a fatal COVID-19 outcome [111].

C3a activates the AKT/β-catenin signaling pathway and can promote the transcription of versican in renal tubular cells. Furthermore, suPAR can bind to tubular cell integrin β6 and activate Rac1, thereby forming versican. In short, C3a and suPAR can result in versican expression in tubular cells, which may lead to renal fibrosis [99]. In experimental models, anti-uPAR monoclonal antibodies have been shown to mitigate these adverse effects in podocytes and tubular cells [85].

### 3.4. suPAR Levels and CKD

An observational study in patients with mild to moderate CKD showed a gradual association of suPAR levels with an increased risk of incident cardiovascular disease [112]. Furthermore, circulating suPAR levels were independently associated with an increased risk of progression to end-stage renal disease (ESRD) in Chinese patients with CKD, especially in patients with pre-existing glomerulonephritis [113]. A clinical study also showed the suPAR level can predict CKD incidence and renal dysfunction progression [114]. The suPAR levels are positively correlated with IgA severity, reflected by pathology findings and subsequent proteinuria [115]. Elevated plasma suPAR levels were also significantly associated with enhanced crescent formation in IgA nephritis [116]. In a large cohort of lupus nephritis patients, elevated suPAR levels were reported to have a higher rate of renal involvement [117].

Elevated plasma suPAR level is strongly associated with incident kidney disease. From a Cardiovascular Biobank analysis, higher suPAR concentrations at baseline are associated with a more prominent deterioration in renal function during the follow-up visits which is also accompanied by a potential for progression to CKD [87]. A study explored the suPAR levels in various kidney pathologies and found minimal change disease has the highest levels and chronic interstitial nephritis has the lowest levels; however, it was elevated in all studied patients with kidney diseases [118]. A meta-analysis revealed that suPAR levels may serve as a prognostic biomarker in CKD patients. The higher suPAR level has been shown to increase the risk of mortality, cardiovascular accidents, and ESRD occurrence [119]. 

Clinically, COVID-19 may pose an increased risk for severe sequelae (e.g., long COVID-19). Patients who have survived COVID-19 are at increased risk for renal outcomes. Therefore, post-acute care for COVID-19 should address kidney disease [120]. In a cohort study of inpatients with AKI, COVID-19-related acute respiratory infection (ARI) was associated with higher rates of post-discharge estimated glomerular filtration rate (eGFR) decline than in patients with AKI without COVID-19 [121]. Thus, we propose that higher levels of suPAR in COVID-19-related AKI may have a higher incidence of long COVID kidney outcomes. However, this possible point of view needs to be further elucidated.

## 4. High Prevalence of Vitamin D Deficiency in COVID-19 Patients

### 4.1. Vitamin D and COVID-19

COVID-19 remains a persistent global health crisis. Type II alveolar epithelial cells develop local innate immunity after the inhalation of SARS-CoV-2. These viral particles may infect circulating macrophages, and T lymphocytes may harbor SARS-CoV-2 antigens. SARS-CoV-2 inhibits diverse immune processes such as pathogen recognition and signaling, the production of IFN, and various interferon-stimulated genes (ISGs) [122]. As various types of T cells are activated and differentiated, the uncontrolled release of cytokines (called a cytokine storm) leads to an exaggerated immune response and tissue destruction. Vitamin D strengthens innate immunity against COVID-19 through activating toll-like receptor 2 and accentuating antimicrobial properties, such as enhancing the synthesis of lysosomes in macrophages, promoting the synthesis and secretion of β-defensins and cathelicidins, and augmenting autophagy through the autophagosomes pathway. Vitamin D also induces adaptive immunity by increasing the population of CD_4_^+^ T cells, activating T cell-dependent B cells to promote virus-specific antibody production, and suppressing T helper 17 cells. Furthermore, vitamin D decreases the release of proinflammatory cytokines by CD_4_^+^ T cells and inhibits the probable exacerbation of a cytokine storm. After SARS-CoV-2 enters cells by binding to the ACE2 receptor with its spike protein, vitamin D improves the bioavailability of soluble ACE2 to capture and inactivate the virus. SARS-CoV-2 activates the RAAS, which is responsible for not only inflammation but also tissue destruction and probable organ failure. Vitamin D inhibits renin expression and acts as a negative regulator of the RAAS [123]. Therefore, vitamin D protects from SARS-CoV-2 by activating intricate mechanisms including both innate and adaptive immunity, enhancing ACE2 expression, and efficiently inhibiting the RAAS. Serum 25-hydroxyvitamin D levels are reported to be negatively associated with the incidence or severity of COVID-19. Although experimental validation is insufficient [53], the clues collected to date probably meet Hill’s criteria for causality in biological systems [124].

The risk factors of COVID-19 include male sex, obesity, hypertension, diabetes, and vitamin D deficiency [125,126]. It is common knowledge that vitamin D deficiency is associated with an increased incidence of microbial infections. Generally, higher risk occurred at 25-hydroxyvitamin D concentrations below 20 ng/mL, but in a retrospective study from the National Health and Nutrition Examination Survey, levels < 30 ng/mL were associated with 58% higher odds of ARI [127,128]. Vitamin D treatment can prevent ARI, especially in patients with severe vitamin D deficiency [129]. In addition, countries with patients having lower average 25-hydroxyvitamin D levels have higher mortality rates from COVID-19 [130]. Patients admitted due to COVID-19 have low levels of 25-hydroxy vitamin D [131]. A meta-analysis study demonstrated that vitamin D deficiency may exacerbate COVID-19 [132]. The administration of high-dose calcidiol or 25-hydroxyvitamin D significantly reduces the need for ICU care in patients hospitalized with COVID-19 [133]. We believe that adequate vitamin D administration is important in attenuating the progression and severity of COVID-19, especially in a population with vitamin D deficiency. 

### 4.2. Vitamin D and suPAR

Vitamin D can inhibit the expression of uPAR in podocytes to prevent damage to podocytes and the subsequent development of proteinuria [23]. In mice with CKD and septicemia, vitamin D inhibits the production of podocyte uPAR, resulting in an anti-proteinuria effect [24]. Vitamin D also attenuated podocyte foot process loss in endotoxemic mice and reduced glomerulosclerosis in nephrectomy rats [24,134]. Amiloride, which can inhibit uPAR synthesis, can be used as a drug to reduce uPAR damage to podocytes in patients with COVID-19 [135]. Due to this inhibitory effect, αVβ3 integrin activation is inhibited and podocyte detachment is reduced [136,137]. Concerning the potential risk of hyperkalemia in AKI, it is not appropriate to administer amiloride instead of vitamin D. Amiloride and vitamin D can inhibit uPAR expression by an off-target mechanism. Whether or not vitamin D can drive uPAR suppression through the vitamin D receptor deserves further evaluation.

### 4.3. Proposed Cross-Talk among Vitamin D, suPAR, and COVID-19-Related AKI

As vitamin D has pleiotropic effects on human pathology [53,138], it can effectively prevent the loss of podocytes, enhance the expression of slit membrane proteins, and maintain the integrity of the renal filtration barrier by attenuating the activity of the RAAS, down-regulating the Wnt/β-catenin signaling, and attenuating the pro-apoptotic pathway [139]. Autophagy is an important protective mechanism of podocytes that is related to angiotensin II (Ang II) levels. Mouse podocytes treated with Ang II exhibited increased calpain activity, inhibited podocyte autophagy, and promoted apoptosis [140]. In cultured human podocytes, Ang II promotes ROS production and podocyte apoptosis by activating ADP-ribosylation factor 6 (Arf6)-Erk1/2-Nox4 signaling [141]. Since COVID-19 patients always have an activation RAAS and a high level of Ag II, this signaling pathway may also occur in COVID-19 patients. Both in vitro and in vivo studies confirmed that Wnt/β-catenin is an important upstream regulator that controls the expression of all RAAS components in the kidneys. The inhibition of Wnt/β-catenin suppresses RAAS activation and ameliorates proteinuria and renal injury [142]. Ang II increasing β-catenin signaling in mouse collection cells leads to increased expression of collagen I, fibronectin, cyclin D1, and c-myc. A reciprocal stimulation relationship between Wnt/β-catenin and Ang II signaling during renal fibrosis seems probable [143] (Figure 4).

In various animal models, vitamin D reduced desmine (a marker of early podocyte lesion), proliferative nuclear cell antigen (PCNA), p27 kinase inhibitor levels, and the local intrarenal activation of the RAAS [139,144,145]. In diabetic nephropathy, vitamin D and its receptor have multiple roles in podocyte, tubulopathy, and interstitial inflammation/fibrosis damage [146]. Though VDR knock-out diabetic mice have developed severe proteinuria and glomerulosclerosis, this study failed to suggest that the action of vitamin D through its receptor and the incidental renoprotective effects may be off-target effects [25]. In doxorubicin-induced podocyte injury, active vitamin D inhibits p38 MAPK signaling and activates the PI3K/AKT survival signaling to prevent podocyte injury [147].

Vitamin D can improve the podocyte damage and proteinuria caused by puromycin aminonucleoside (PAN); this beneficial effect may be due to the direct regulation of TGF-β1/BMP-7 signaling [148]. The PAN-induced podocyte damage can also be ameliorated by treating it with the Smad3 inhibitor SIS3. Since the expression of the actin-binding protein transgelin is induced by the Smad3 signaling pathway during the PAN-induced podocyte lesion, it can be reversed by blocking this pathway [149]. The effects of vitamin D through the nephrin signaling pathway were demonstrated by assessing the expression of podocalyxin and nephrin in isolated rat glomeruli. Previous studies also showed that active vitamin D treatment can enhance the vitamin D receptor (VDR) expression, which will facilitate the co-localization of VDR and RXR (retinoid X receptor) in the nucleus, resulting in the increased expression of nephrin, which promotes glomerular survival [150] (Figure 4).

Vitamin D is an inhibitor of renin biosynthesis and a modulator of RAAS activity; therefore, VDR dysregulation will increase renin and angiotensin II levels, which in turn leads to hypertension and cardiac hypertrophy [151,152,153]. In addition, VDR is involved in controlling the degree of inflammation, epithelial-mesenchymal transition, and maintaining podocyte integrity [154,155]. Vitamin D can effectively suppress kidney inflammation thanks to VDR sequestration of NF-κB signaling [156]. Vitamin D also inhibits NF-κB transaction by modulating advanced glycation end products (AGE) and receptors for AGE (AGE-RAGE system) [157]. In vitamin D deficiency, vitamin D supplementation can lower AGE levels and increase soluble RAGE levels [158].

Vitamin D helped in protecting mitochondrial morphology in kidney tissue by reversing high AT1 receptor expression and low Hsp70 expression in spontaneously hypertensive rats [159]. Mitochondrial preservation underlies the cellular functions of Ca^2+^ signaling regulation, ROS status regulation, adenosine triphosphate production, and redox reactions [151,160]. VDR activation also promotes mitochondrial integrity through the ligand-independent control of the permeability transition pore [161].

## 5. AKI Management in COVID-19

### 5.1. Targeting AKI in COVID-19

Several experimental drugs are currently being developed as anti-viral treatments for COVID-19, but their effects on AKI are unknown [162]. There are currently no specific treatment options for COVID-19-related AKI, so intensive care is largely supportive. At present, approaches to managing AKI and identifying potential indications for RRT are largely based on non-COVID-19 clinical experience. Furthermore, AKI strategies are empirically tailored to patients with COVID-19 [163].

### 5.2. Adjunctive Therapy with Vitamin D

It is unclear whether recovery from COVID-19-related AKI differs from that of other forms of AKI. Further studies are needed to better understand the direct effects of SARS-CoV-2 on long-term renal injury and recovery [34]. Some patients with COVID-19 develop pulmonary fibrosis after recovery [164]. While we do not know whether progressive renal fibrosis develops in patients recovering from COVID-19-related AKI, the development of fibrosis and progression to CKD in patients recovering from other forms of AKI suggests that this should likely occur. Patients with COVID-19-related AKI must be monitored for a minimum of two to three months or more after discharge to assess renal recovery [165]. Vitamin D attenuates systemic inflammation and thus decreases circulating myeloid-cell-derived suPAR, resulting in the amelioration of kidney tubule, glomerulus, and podocyte injury. Furthermore, vitamin D attenuates the local renal production of uPAR (at least in podocytes) and preserves the glomerular barrier function and structure, responsible for maintaining adequate renal function. Furthermore, vitamin D may exhibit a powerful influence on podocytes by inhibiting apoptosis and structural fibrosis [34].

Recent association studies explore the possible role of vitamin D deficiency in COVID-19. Low levels of 25(OH)D may be a result of COVID-19 [166]. Therapeutic doses of vitamin D supplements may not be dangerous for people with COVID-19, and may even reduce disease progression or severity. Given the urgency of the current situation, sufficient evidence to support this claim remains to be proven. The recommended intake of 10,000 IU/day for 7–14 days, followed by 5000 IU per day for individuals at high risk of COVID-19 to achieve target blood 25(OH)D concentrations of over 40 to 60 ng/mL has been suggested [167].

## 6. Conclusions

Current evidence suggests that high levels of suPAR are probably a risk factor for AKI in patients with COVID-19 [18] and this AKI may be related to vitamin D status [168]. Clinically, higher levels of suPAR are linked to a rapid deterioration of renal function in children with CKD [169]. Measuring suPAR levels in patients with COVID-19 may be a crucial tool for improving disease severity accuracy and predicting complications of SARS-CoV-2 infection, especially in comorbid AKI and micro- and macro-thrombotic events. Increased suPAR concentrations increase oxidative stress from mitochondrial and extra-mitochondrial enzymes in renal cells, [170,171], which synergizes with direct SARS-CoV-2 renal injury and indirect micro-thrombosis in the renal vasculature, therefore increasing the risk and severity of AKI [172]. Elevated circulating suPAR levels may decrease active plasmin production via the competitive inhibition of cell membrane uPA [173]. Ultimately, the development of hypercoagulability in gravely ill COVID-19 patients leads to severe thrombosis [10,174,175].

Vitamin D deficiency occurs frequently in patients with COVID-19 [176]. Patients with COVID-19-related AKI have increased consumption and reduced kidney capacity to produce vitamin D, which may aggravate the vitamin D insufficiency. In these patients, all have increased inflammatory cytokine levels, which can accentuate local and systemic suPAR levels. Vitamin-D-deficiency-induced inflammation further promotes suPAR production from myeloid cells and podocytes. A high level of suPAR causes damage to kidney cells such as tubular cells, podocytes, endothelial cells, and MCs. High suPAR levels damage renal tubular cells, which further decreases vitamin D production. The administration of vitamin D inhibits local and systemic suPAR production by reducing the severity of inflammation [53,176], protects podocytes, tubular cells, and MCs, and prevents microthrombus formation in the glomerular capillaries and other vascular structures [35]. Based on this basic and clinical evidence, here we propose a possible link between vitamin D, suPAR, and COVID-19-related AKI. Indeed, the mechanisms underlying these hypotheses require further study (Figure 5). Given the well-established role of vitamin D in COVID-19, among high-risk SARS-CoV-2-positive patients, individuals and older adults with comorbidities are expected to receive vitamin D supplementation during the COVID-19 pandemic [177]. Moreover, vitamin D is comparatively inexpensive, safe, and widely available. Therefore, supplementation could be considered in these high-risk groups, especially in patients with AKI [53,168]. However, well-designed research is needed to better explore the possible linkage between vitamin D deficiency and COVID-19 morbidity and mortality, and to assess whether the correction of this deficiency can avoid or lessen the severity of COVID-19-related AKI. 

## Figures and Tables

**Figure 1 ijms-23-10725-f001:**
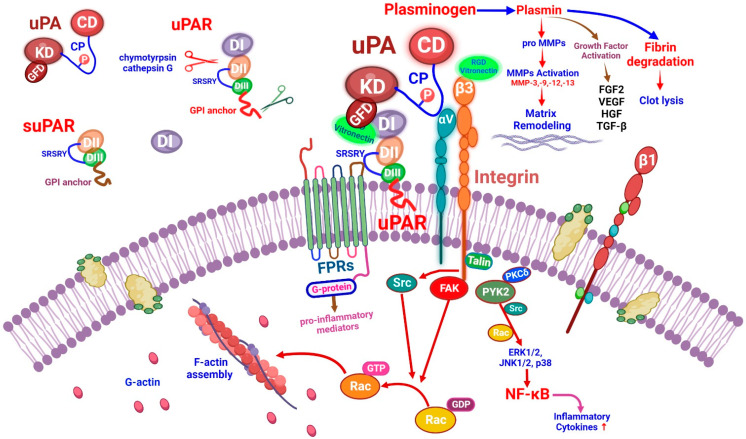
The urokinase plasminogen activator (uPA)/uPA receptor (uPAR) system function. The uPAR lacks an intracellular domain, however, it can interact with other transmembrane receptors such as formyl peptide receptors (FPRs) and integrins (mainly, αvβ3 integrin). The activation of these co-receptors then triggers intracellular signaling.

**Figure 2 ijms-23-10725-f002:**
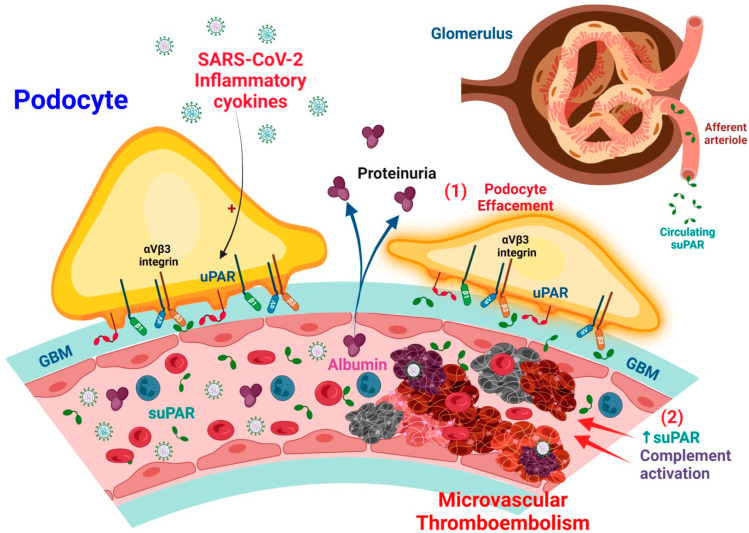
Putative molecular mechanism of suPAR-related podocyte injury and glomerular capillary microthrombus formation in COVID-19-related AKI. (1) High-level suPAR activation of integrin αvβ3 signaling impairs podocytes. (2) High levels of suPAR in combination with complement activation exacerbate the hypercoagulable state that leads to the development of AKI.

**Figure 3 ijms-23-10725-f003:**
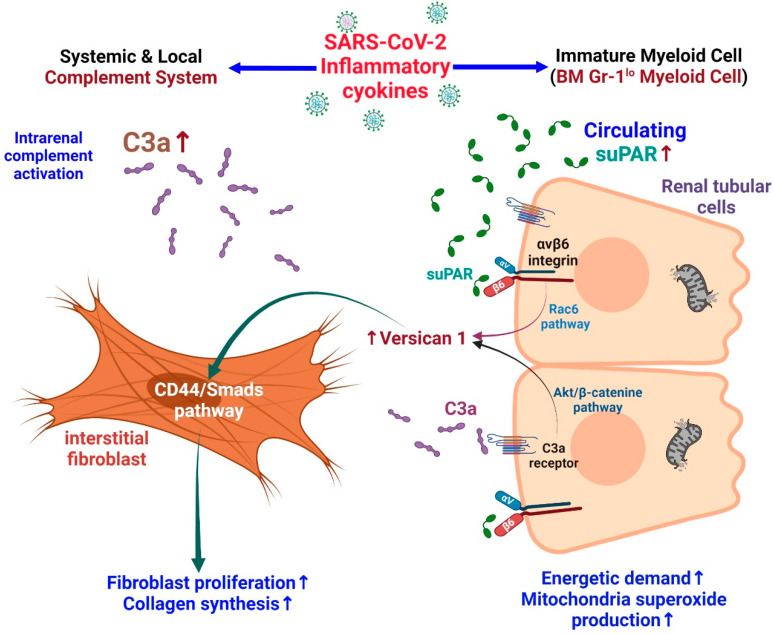
SARS-CoV-2-infection-induced injury to the renal tubular cells. Patients with severe SARS-CoV-2 infection are more likely to have high suPAR levels. In renal tubular cells, suPAR can bind to integrin β6 and activate Rac1 leading to the formation of versican V1. suPAR also affects proximal tubular cells, altering mitochondrial respiration and inducing oxidative stress. Patients with severe SARS-CoV-2 infection are also more likely to use C3. The increased generation of C3a in severe COVID-19 is accompanied by the release of activated CD16^+^ cytotoxic T cells. C3a promotes the transcription of versican by activating the AKT/β-catenin pathway. Tubular cell-derived versican V1 induces proliferation and collagen synthesis by activating the CD44/Smad3 pathway in fibroblasts. Briefly, C3a and suPAR can drive versican V1 expression in tubular cells, which contributes to fibrosis of renal interstitial tissues.

**Figure 4 ijms-23-10725-f004:**
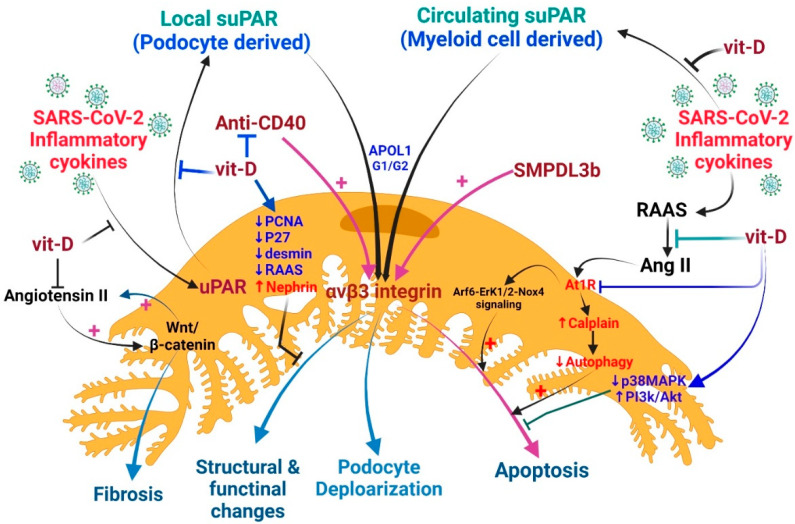
Potential mechanisms by which vitamin D protects podocytes and reduces proteinuria in COVID-19-related AKI. The SARS-CoV-2 infection causes renal podocytes to overproduce suPAR locally and systemically; suPAR activates αvβ3 integrins in podocytes and leads to apoptosis, depolarization, and structural and functional changes in podocytes. At this time, anti-CD40, SMPDL3, and G1/G2 with APOL1 significantly increase the activation of αvβ3 integrins and damage podocytes. Treatment with vitamin D may not only reduce systemic inflammation but also reduce BM-derived leukocyte suPAR. Vitamin D may also reduce the production of uPAR in podocyte cells and protect podocyte function by reducing the deleterious effect of angiotensin II and AT1R signaling.

**Figure 5 ijms-23-10725-f005:**
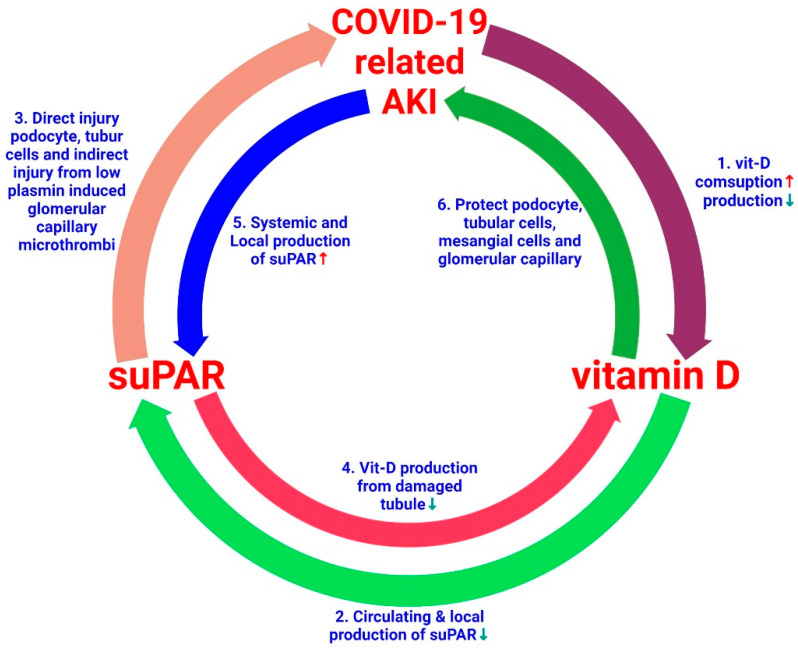
A putative cross-talk between vitamin D, suPAR, and COVID-19-related AKI. (1) Patients with COVID-19-related AKI have increased consumption and reduced kidney capacity to produce vitamin D, which may lead to vitamin D deficiency. (2) Vitamin D insufficiency or deficiency increases suPAR production from myeloid cells and podocytes. The addition of vitamin D suppresses local and systemic suPAR levels. (3) A high level of suPAR causes various lesions in the kidney cells such as tubular cells, podocytes, endothelial cells, and mesangial cells. (4) High levels of suPAR damage the renal tubular cells, resulting in a decrease in vitamin D production. (5) Patients with COVID-19-related AKI may have increased systemic and local suPAR levels. (6) Vitamin D supplementation can protect podocytes, tubular cells, and mesangial cells and prevent microthrombus formation in the glomerular capillaries and other vascular structures.

## Data Availability

This is a narrative review article. The primary collection of documents for analysis and review comes from PubMed.

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
