# Peer review of "The Perspective of Vitamin D on suPAR-Related AKI in COVID-19"

_ijms, 2022, doi:10.3390/ijms231810725_

Round 1
Reviewer 1 Report (Previous Reviewer 2)
The authors are to be commended for making such extensive changes in a way that has improved the readability of the paper. A few minor comments remain:
Line 298-299: Please describe the impact of bacterial infection in the context of pneumonia infections after or during COVID-19 (e.g., "Post-COVID pneumonia") which has been described in the clinical literature.
Line 313-334: This paragraph isn't really needed in the context of COVID.
Lines 434-442 and 520-531: Please indicate the actual blood test levels of Vit. D that indicate deficiency so doctors can follow clinical guidelines given later on in the paper for replenishment.
Author Response
Response to Reviewer 1
The authors are to be commended for making such extensive changes that have improved the paper's readability. A few minor comments remain:
Q1: Line 298-299: Please describe the impact of bacterial infection in the context of pneumonia infections after or during COVID-19 (e.g., "Post-COVID pneumonia") which has been described in the clinical literature.
Response: Thank you very much for the reviewer’s valuable comments. We have added some descriptions about the pneumonia infection after/during COVID-19.
Severe or critical COVID-19 is associated with an increased secondary infection rate and would lead to a significantly worsened prognosis. Secondary infection risks increased after receiving invasive respiratory ventilation and intravascular devices. The most common infections were respiratory and the most detected pathogens were gram-negative bacteria, followed by gram-positive bacteria, viruses, and fungi [PMID: 32815458]. (Line 304~)
Q2: Line 313-334: This paragraph isn't really needed in the context of COVID.
Response: Thank you very much for the reviewer’s crucial suggestions. This paragraph discusses the effects of suPAR on kidney podocyte injury. Since suPAR will significantly influence glomerular barrier function in COVID-19 patients. So we would like to preserve this paragraph so the reader to easily understand how the suPAR may disturb podocyte function and viability. We hope that the reviewers agree to keep this paragraph.
Q3: Lines 434-442 and 520-531: Please indicate the actual blood test levels of Vit-D that indicates definition doctors can follow clinical guidelines given later on in the paper for replenishment.
Response: Thank you very much for the reviewer’s important comments.
Generally, higher risk occurred at 25-hydroxyvitamin D concentrations below 20 ng/mL, but in a retrospective study from the National Health and Nutrition Examination Survey, levels <30 ng/mL were associated with 58% higher odds of acute respiratory infections [PMID: 23677871, PMID: 25903964]. (Line 446~)

Reviewer 2 Report (New Reviewer)
A valuable work, a well-chosen path of interdependencies: AKI, COVID-19 and vitamin D. Great introduction, clearly justifies the need to work out the topic. The rich bibliography should be emphasized, including over half of the publications from the last 5 years.
The following comments are intended to indicate points where the work should be improved in order to improve its reception and quality:
- Figures 1, 2 and 4 are not a full graphic representations of the text. Figure 1 - some of the abbreviations / names used in the description of the Figure do not appear in the text, such as: CD, KD, CP, PYK2, GFD or included in the text are not in the description of the Figure (vitronectin). Figure 2 - lack of indication of interdependencies by using, for example, appropriately directed arrows. Figure 4 - the description in the text between lines 506-510 is not depicted in the Figure;
- subchapter 3.3.3 - the authors do not emphasize the participation of suPAR in this section, which is not consistent with its title: "Effects of suPAR on mesangial cell (MC) injury";
- lines 504-505: no reference to a literature source;
-all abbreviations used in the text should be explained, with the principle of explanation at first appearance - no explanations in the text, eg for: GPI (line 69); DAMP, PAMP (line 180); FSGS (line 340); GBM (line 362); ESRD (line 402); eGRF (line 406); ARI (line 406);
- correction of typing errors requires Figure 4 and line 451: Amilorid, not: Amolorid.
Author Response
Response to Reviewer 2
A valuable work, a well-chosen path of interdependencies: AKI, COVID-19 and vitamin D. Great introduction, clearly justifies the need to work out the topic. The rich bibliography should be emphasized, including over half of the publications from the last 5 years.
The following comments are intended to indicate points where the work should be improved in order to improve its reception and quality:
- Figures 1, 2 and 4 are not a full graphic representations of the text.
Q1: Figure 1 - some of the abbreviations / names used in the description of the Figure do not appear in the text, such as: CD, KD, CP, PYK2, GFD or included in the text are not in the description of the Figure (vitronectin).
Response: We have corrected these mistakes according to the reviewers’ comments. (Line 214~, 235~). The vitronectin is already present in our original figure 1. We change the color of the vitronectin in figure 1 to easily arouse the reader’s attention (Figure 1). (Line 247)
Q2: Figure 2 - lack of indication of interdependencies by using, for example, appropriately directed arrows.
Response: We have made a necessary rectification according to the reviewers’ suggestions. The figure and legends were categorized into two subtopics including (1) High-level suPAR activation of integrin αvβ3 signaling impair podocytes. (2) High levels of suPAR in combination with complement activation exacerbate the hypercoagulable state that leads to the development of AKI. (Line 291~)
Q3: Figure 4 - the description in the text between lines 506-510 is not depicted in the Figure;
Response: We delete Figure 4 in line 521 due to the crowded figure contents. (Line 523)
Q4: - subchapter 3.3.3 - the authors do not emphasize the participation of suPAR in this section, which is not consistent with its title: "Effects of suPAR on mesangial cell (MC) injury";
Response: We have made a necessary rectification according to the reviewers’ crucial comments. We change the subtitle to “The αvβ3 integrin-dependent activation and mesangial fibrosis”. (Line 367)
Q5: - lines 504-505: no reference to a literature source;
Response: We have added the necessary reference. Reference 159. (Line 518)
Q6: -all abbreviations used in the text should be explained, with the principle of explanation at first appearance - no explanations in the text, eg for: GPI (line 69); DAMP, PAMP (line 180); FSGS (line 340); GBM (line 362); ESRD (line 402); eGRF (line 406); ARI (line 406);
Response: We have corrected all abbreviation statements.
Q7: - correction of typing errors requires Figure 4 and line 451: Amilorid, not: Amolorid.
Response: We had corrected it. (Line 464)

This manuscript is a resubmission of an earlier submission. The following is a list of the peer review reports and author responses from that submission.
Round 1
Reviewer 1 Report
The proposed links between vitamin D and suPAR, and the consequent effect on kidney function in COVID-19 is highly speculative. The authors have presented very little convincing evidence of the links. Most of the scientifically robust information is confirming that (i) suPAR could serve as a marker of clinical severity of COVID-19, which has been well-established and (ii) vitamin D deficiency can potentially worsen prognosis in COVID-19.
However, it is not very clear how the authors have combined these facts to account for AKI in patients with COVID-19. Only one or two animal studies are cited as perhaps indicating that vitamin D might suppress suPAR levels. This is scientifically insufficient to draw the conclusions that the authors have. Also, there is no evidence that suPAR is directly involved in causing AKI.
Unreferenced statements such as “Current evidence shows that increased suPAR level is a risk factor for developing AKI in patients with COVID-19 and this is related to vitamin D status” and “The administration of vitamin D suppresses local and systemic suPAR 626 production, protects podocytes, tubular cells, and MCs, and prevents microthrombus 627 formation in the glomerular capillaries and other vascular structures” are simply incorrect and unsubstantiated. The title of the manuscript is also misleading in this regard; no information on the therapeutic effects of vitamin D on suPAR levels in COVID-19-related AKI is presented. It is all very theoretical and highly speculative. Internationally, there isn't enough data to recommend use of vitamin D to prevent infection with the virus that causes COVID-19 or to treat COVID-19 (according to both the US National Institutes of Health and the World Health Organization). The lack of a consistent protective signal from RCTs reporting so far is reflected by the absence of any recommendation relating to prophylactic or therapeutic use of vitamin D for COVID-19 in guidelines from national or international bodies (Martineau AR, Cantorna MT. Vitamin D for COVID-19: where are we now? Nat Rev Immunol 2022 https://doi.org/10.1038/s41577-022-00765-6).
The manuscript would be better submitted to a medical hypotheses journal, with the title amended to reflect that it is theoretical.
Author Response
Response to reviewer 1:
Comments and Suggestions for Authors:
Question: The proposed links between vitamin D and suPAR, and the consequent effect on kidney function in COVID-19 is highly speculative. The authors have presented very little convincing evidence of the links. Most of the scientifically robust information is confirming that (i) suPAR could serve as a marker of clinical severity of COVID-19, which has been well-established and (ii) vitamin D deficiency can potentially worsen prognosis in COVID-19.
Response: I thank the reviewers for their time and invaluable contribution.
In this review article, we attempt to link suPAR in vitamin D treatment to COVID-19-related AKI. This may provide another window for potential therapeutic consideration of vitamin D in the suPAR-mediated AKI for COVID-19.
Question: However, it is not very clear how the authors have combined these facts to account for AKI in patients with COVID-19. Only one or two animal studies are cited as perhaps indicating that vitamin D might suppress suPAR levels. This is scientifically insufficient to draw the conclusions that the authors have. Also, there is no evidence that suPAR is directly involved in causing AKI.
Response: Thank you for the reviewer’s valuable comments.
We agree with the reviewer’s suggestion. Our previous study showed COVID-19 patients have a high incidence of kidney injury. Beyond the cytokine storm and hemodynamic instability, SARS-CoV-2 might directly induce kidney injury and cause histopathologic characteristics, including acute tubular necrosis, podocytopathy, and microangiopathy (J Clin Med. 2020 Nov 3;9(11):3547, J Pers Med. 2021 May 19;11(5):432 ). Another of our papers also showed that AKI is one of the most common extrapulmonary manifestations of severe COVID-19. There are some correlations between vitamin D levels and COVID-19 severity through multiple pathways (Int J Mol Sci. 2022 Jul 1;23(13):7368). Elevated serum suPAR concentration is a negative prognostic indicator for several severe clinical diseases including AKI (Biochim Biophys Acta Mol Basis Dis. 2021 Oct 1;1867(10):166186). A previous animal study showed that vitamin D inhibits podocyte uPAR expression in vitro and in vivo, which may be an unanticipated off-target effect of vitamin D and explains its antiproteinuric effect (PLoS One. 2013 May 31;8(5):e64912) (Line 543). In addition, clinical findings of admission suPAR levels in patients hospitalized with COVID-19 predict in-hospital AKI and the need for dialysis. The suPAR may be a key component of AKI pathophysiology in COVID-19 (J Am Soc Nephrol. 2020 Nov;31(11):2725-2735). Therefore, we propose this narrative presumptive review article. For convincing the readers, we would like to change our title to “Potential beneficial effects of vitamin D on suPAR-mediated AKI in COVID-19”
Question: Unreferenced statements such as “Current evidence shows that increased suPAR level is a risk factor for developing AKI in patients with COVID-19 and this is related to vitamin D status” and “The administration of vitamin D suppresses local and systemic suPAR production, protects podocytes, tubular cells, and MCs, and prevents microthrombus formation in the glomerular capillaries and other vascular structures” are simply incorrect and unsubstantiated. The title of the manuscript is also misleading in this regard; no information on the therapeutic effects of vitamin D on suPAR levels in COVID-19-related AKI is presented. It is all very theoretical and highly speculative. Internationally, there isn't enough data to recommend use of vitamin D to prevent infection with the virus that causes COVID-19 or to treat COVID-19 (according to both the US National Institutes of Health and the World Health Organization). The lack of a consistent protective signal from RCTs reporting so far is reflected by the absence of any recommendation relating to the prophylactic or therapeutic use of vitamin D for COVID-19 in guidelines from national or international bodies (Martineau AR, Cantorna MT. Vitamin D for COVID-19: where are we now? Nat Rev Immunol 2022 http://sci-hub.tw/10.1038/s41577-022-00765-6).
Response: Thanks to the reviewers for the important suggestions. We have all learned a lot from the reviewers' criticisms.
To convince our readers, we would like to change our title to “Potential beneficial effects of vitamin D on suPAR-mediated AKI in COVID-19”.
We correct our statements based on the reviewers' comments, e.g.
“Current evidence shows that increased suPAR level is a risk factor for developing AKI in patients with COVID-19 and this is related to vitamin D status” will be changed to
” Current evidence suggests that elevated suPAR levels may be a potential risk factor for AKI in COVID-19 patients (J Am Soc Nephrol. 2020 Nov;31(11):2725-2735) and this AKI may be related to vitamin D status (Int J Mol Sci. 2022 Jul 1;23(13):7368)”. (Line 636~637)
The sentence
“The administration of vitamin D suppresses local and systemic suPAR production, protects podocytes, tubular cells, and MCs, and prevents microthrombus formation in the glomerular capillaries and other vascular structures” will be changed to
“Administration of vitamin D inhibits local and systemic suPAR production by reducing the severity of inflammation (Int J Mol Sci. 2021 May 16;22(10):5251, Int J Mol Sci. 2021 Aug 20;22(16):8988.), protects podocytes, tubular cells, and MCs, and prevents microthrombus formation in the glomerular capillaries and other vascular structures (J Clin Med. 2020 Nov 3;9(11):3547)”. (Line 659~662)
Questions: The viewpoint of the reviewer about the lack of a consistent protective signal from RCTs reporting so far is reflected by the absence of any recommendation relating to the prophylactic or therapeutic use of vitamin D for COVID-19 in guidelines from national or international bodies.
Response: We would like to make some explanations.
Vitamin D deficiency (VDD) occurs all over the world, predominantly in the Middle East, China, Mongolia, and India. In these areas, the mean serum 25(OH)D level is lower and the percentage of VDD is higher than those in Europe or America (Endocrinol Metab Clin North Am. 2017;46:845–870). In general, there is a large gap between officially recommended dietary reference intakes of vitamin D and the high prevalence of vitamin D deficiency in the general population, requiring action by health authorities. However, systemic vitamin D food fortification is an effective approach to improve vitamin D status in the general population and has been used in countries such as the United States, Canada, India, and Finland (Front Endocrinol (Lausanne). 2018 Jul 17;9:373). Because food fortification policies increase basal serum vitamin D levels, a large number of recent studies in US or Canadian populations have shown no apparent beneficial effect on further vitamin D supplementation. Our recent editorial article also explains the impact of systemic vitamin D food fortification on vitamin D deficiency, which can impact the efficacy of a COVID-19 vaccine (Eur J Intern Med. 2021 Nov;93:114). We suggest that avoiding vitamin D deficiency may have beneficial effects on COVID-19 patients, especially in populations living in areas without dietary vitamin D fortification.

Reviewer 2 Report
Verdict: Major Revisions
Please see attached PDF for details

Author Response
Reviewer 2
General Comments:
Question: The review is well written from a grammatical standpoint and the figures are a welcome way to grasp complex molecular concepts. However, the paper must focus on COVID, Vitamin D and suPAR. Instead, it attempts to add every bit of knowledge of every possible kidney infection or disease and link it somehow with COVID and suPAR. This paper would be stronger if the non-COVID, non-suPAR, and non-Vitamin D paragraphs were trimmed considerably. The sections do read like they were written by different people who didn’t read the other sections and has some organizational problems. In general, this topic is important and, if the paper is substantially reorganized, it will provide a valuable resource for clinicians.
Response: Thanks a lot for the reviewer's suggestions.
We have significantly reduced paragraphs not related to COVID, not related to suPAR, and not related to vitamin D. We reorganized this manuscript based on feedback from critics.
Specific Comments:
Introduction:
- Line 79-87: You must indicate some mechanism for suPAR regulation by Vitamin D. Be specific with regard to genes, proteins, and regulatory elements involved. Vitamin D is a hormone and has potent direct regulatory effects; in case there is no direct evidence, please provide a proposed mechanism. It can be just a couple of sentences but will set up the rest of the paper well.
Response: Thank you very much for your critical comments.
Previous animal study demonstrated that 1,25(OH)2D3 inhibited podocyte uPAR expression in vitro and in vivo, which may be an unanticipated off-target effect of vitamin D and explain its anti-proteinuric effect in the 5/6 nephrectomy rat FSGS model and the LPS (Lipopolysaccharides) mouse model of transient proteinuria (PMID: 23741418). Although diabetic vitamin D receptor knockout mice developed more severe proteinuria and glomerulosclerosis due to increased glomerular basement membrane thickening and podocyte effacement (PMID: 17928826). Combining these findings, we propose that the effect of vitamin D on uPAR cannot be shown to be through the vitamin D receptor and that this renoprotective effect may be an off-target effect (PMID: 23741418). In the absence of direct evidence, we hypothesized that the protective effect of vitamin D may be directly through attenuating uPAR/suPAR-related integrin signaling and indirect anti-inflammatory pathways. However, the molecular mechanism by which vitamin D inhibits podocyte uPAR in COVID-19-related AKI remains to be fully elucidated. (Line 98~106)
- What is the difference between uPAR and suPAR and why would there be an insoluble and soluble version of this protein? Although you do describe it later, the introduction is a good place to establish the basics. Be specific when describing exactly what suPAR is. Since this is the key protein of your paper, also distinguish its action and effects from plasminogen activator inhibitor-1 (PAI-1).
Response: We thank you for the reviewer’s valuable comments.
We have made some description on the uPAR and suPAR. The uPA and its corresponding receptor (uPAR or CD87) form a multimolecular complex at the cell surface which can cause fibrinolysis and perform immune functions. The uPAR may be detached from the cellular membrane GPI anchorage, resulting in the release of the soluble form of suPAR. The uPA localized in podocytes/tubular cells can bind to the uPAR domain, then the uPAR can interact with integrins and vitronectin. Damaged endothelial cells produce and secrete plasminogen activator-inhibitor 1 (PAI-1), the binding of PAI-1 to the uPA-uPAR complex can inhibit the activation of plasminogen. Following its binding to uPAR, uPA catalyses the conversion of plasminogen to plasmin, a serum protease involved in the degradation of the extracellular matrix (ECM) and cell motility. Once uPA binds to uPAR, a conformational transition occurs and may play a role in linking with other co-receptors such as integrins and vitronectin. The uPAR act on αvβ3 integrins can stimulate cell motility and invasion in kidney podocytes. The binding of the vitronectin with uPAR and β3 integrals is necessary for the activation of catabolic proteases such as MMP-3 and MMP-13. (Line 64~78).
- In this section, COVID-19 is used then Section 2 switches to SARS-CoV-2. Please choose a term and use it consistently.
Response: We thank you for the reviewer’s important suggestions.
In this manuscript, COVID-19 represents disease and SARS-CoV-2 represents virus. We have carefully examined both nouns in the revised manuscript.
Section 2
- Line 94: What is KDIGO?
Response: We thank you for the reviewer’s valuable comments.
Kidney Disease: Improving Global Outcomes (KDIGO) (Line 116).
- Line 107: Lower recovery rates of what are expected? Overall recovery or kidney-specific functional recovery? (Section 2.1)
Response: We thank you for the reviewer’s valuable suggestions.
It means lower renal recovery rates are expected. (Line 130).
- Line 111-112: Please eliminate any non-COVID-19-specific data unless it can be used in context to determine why this particular group suffers more from AKI after COVID-19. In this case, you might want to discuss any specific co-morbidities that the group may suffer from that could increase the likelihood of AKI after COVID. (Section 2.1)
Response: We thank you for the reviewer’s valuable comments.
We erase the non-COVID-19-specific data as reviewer’s suggestion. We add the statement that the most common risk factors related to SARS-Cov-2-mediated AKI include diabetes, obesity or hypertension and previous chronic kidney disease (CKD) [PMID: 33060844]. (Line 133~135)
- Line 162-163: What does CKD activation of RAAS have to do with COVID? If you are stating that CKD patients who contract COVID have kidney damage, this is an obvious conclusion. Please offer some proposed mechanisms by which COVID can activate RAAS to synergistically increase blood pressure and damage the kidneys.
Response: We thank you for the reviewer’s crucial opinions. (Section 2.2.1)
The activation of the renin–angiotensin–aldosterone system (RAS) and Ang-II-related inflammation and fibrosis play vital roles in COVID-19 infection and mortality. After SARS-CoV-2 entry into the host cell, the virus downregulates ACE2 expression, which in turn upregulates angiotensin II (Ang II) (PMID: 32504757). In addition to elevated blood pressure, Ang II interacts with its receptor, Ang II receptor type 1, and modulates the gene expression of several inflammatory cytokines via nuclear factor κB signaling. This interaction also promotes macrophage activation and results in the production of inflammatory cytokines that may cause acute respiratory distress syndrome or macrophage activation syndrome (PMID: 34065735). (Line 191~197)
- Line 195-196: Please be specific here. Is it the left ventricle or atrium? The right ventricle or atrium?
Response: We thank you for the reviewer’s valuable suggestions. (Section 2.2.2)
An increase in right ventricle pressure can result from left ventricle failure or high positive expiratory pressure on mechanical ventilation in patients with respiratory failure. (Line 217~218)
- Line: 197-198: Why would ER stress play a critical role and how would this occur in the “indirect” method proposed?
Response: We thank you for the reviewer’s valuable comments. (Section 2.2.2)
Mitochondrial dysfunction induced by SARS-CoV-2 can damage kidney cells directly and indirectly by enhancing systemic oxidative stress and inflammation. (Line 219~221)
- Line 200-202: This directly contradicts the statement in Lines 191-192. Suggest trimming Section 2.2.2 down to 1 short paragraph as much of it is not relevant directly to the suPAR/COVID interaction that the paper is based around.
Response: We thank you for the reviewer’s valuable comments.
In one report, platelet counts remain normal in most patients with disseminated intravascular coagulation; this suggests that the kidneys are not seriously damaged by thrombotic microangiopathy. This suggest beyond thrombotic microangiopathy, there are other crucial factors may influence the kidney injury. To avoid confusing readers, we have removed this sentence. By the way, we also reduced Section 2.2.2 to 1 short paragraph.
Section 3
- Line 209-210: Again, AKI outside of the COVID effect on kidneys is not relevant unless this paper is focused on the effect of COVID on patients with pre-existing kidney disease.
Response: We thank you for the reviewer’s important suggestions.
We have deleted this sentence.
- Because English has far too many rules for no good reason, you can never start a sentence with a lowercase letter, even if it’s a part of the official name. Suggest changing Line 217 to “The uPA molecule and its corresponding…”
Response: We thank you for the reviewer’s critical comments.
We correct this mistake. (Line 234)
- Line 227-228: What effect does the “powerful driver” of the transition have on COVID or the immune system or inflammation in the kidney?
Response: We thank you for the reviewer’s valuable comments.
It means the active TGF-β is a powerful driver of endothelial-mesenchymal transition in kidney. (Line 242)
- Figure 1: This caption needs to be part of the main text. A more suitable caption would focus on the part of the uPA/suPAR pathway that directly deals with either Vitamin D regulation or COVID or both.
Response: We thank you for the reviewer’s valuable comments.
We have moved most of the legend of Figure 1 into the main text of Section 3.1. We discuss the basic uPA/suPAR pathway in Figure 1. The vitamin D-related uPA/suPAR pathway is presented in Figure 4.
- Line 256: Suggest changing to “…this pathway. This uPAR-B3 integrin…”
Response: We thank you for the reviewer’s valuable comments.
We have corrected it as reviewer’s suggestion. (Line 271)
- Lines 262-278: This is a list of integrins but how they function during COVID infection is more relevant and so Lines 269-270 should be clarified to present evidence that these integrins are linked to renal damage during or after COVID. If they are not yet linked by experimental evidence, then a proposed mechanism is welcome here.
Response: We thank you for the reviewer’s valuable comments.
Since patients with COVID-19-related AKI may have high serum suPAR levels, we speculate that these high suPARs can activate integrin signaling that may cause kidney injury. (Line 299~301)
- Line 271-272: If this is true, it needs to be supported by evidence in COVID patients in Section 2 as excess collagen and ROS would be chief agents to damage the delicate vascular network of the kidneys.
Response: We thank you for the reviewer’s valuable comments.
These sentences are used to introduce the collagen receptors in the kidney, integrin α1β1 and α2β1, which play an important pathogenic role in inflammatory forms of kidney damage. However, uPA/uPAR mainly acts on the αvβ3 integrin. Since COVID-19-related AKI always has high levels of suPAR, thus it may also play a role in the pathogenesis of kidney injury. (Line 293~295)
- Figure 2: This also needs to be a main section of text and use a shortened caption for the actual figure.
Response: We thank you for the reviewer’s valuable comments.
We have moved most of the legend of Figure 2 to the main text of new Section 3.2.
- Line 284: Change to “…lead to AKI. In this case, suPAR is produced…”
Response: We thank you for the reviewer’s valuable comments. We have corrected this mistake. (Line 305)
- Lines 305 and 307: Please find a way to start these sentences with uppercase letters, e.g., “converts plasminogen into plasmin; uPA itself is fibrin independent…” Please continue to fix this error throughout the manuscript.
Response: We thank you for the reviewer’s valuable comments. We have corrected these mistakes. (Line 330, 332)
- Any discussion of PAI-1 and uPA release upon infection must be focused on COVID, as viral infections trigger a different immune response since bacterial LPS/TLR4 pathways are not responsive to viruses. Likewise, Line 321-322 is superfluous unless presented in the context of co-morbid COVID and fungal infections.
Response: We thank you for the reviewer’s valuable comments. The phrase has been deleted to avoid confusing readers.
- Sections 3.2.4 and 3.3 are good. Section 3.3 needs to focus on how suPAR levels exacerbated by COVID can result in long-term CKD, especially with regard to the “long COVID” phenomenon. (New 3.4)
Response: We thank you for the reviewer’s valuable comments.
Clinically, COVID-19 is associated with increased risk of post-acute sequelae involving pulmonary and extrapulmonary organ systems-referred to as long COVID. Patients who survived COVID-19 exhibited increased risk of kidney outcomes in the post-acute phase of the disease. Post-acute COVID-19 care should include attention to kidney disease (PMID: 34470828). In this cohort study of US patients who experienced in-hospital AKI, COVID-19-associated AKI was associated with a greater rate of eGFR decrease after discharge compared with AKI in patients without COVID-19, independent of underlying comorbidities or AKI severity (PMID: 33688965). Thus, we propose that higher levels of suPAR in COVID-19-related AKI may have a higher incidence of long COVID kidney outcomes. However, this speculation need further elucidation. (Line 469~478)
Section 4
- Section 4.1 is good but how much Vitamin D is protective? What are the clinical guidelines for physicians who want to supplement Vitamin D? What is the best form and dose? Does COVID-19 deplete Vitamin D levels or are high levels persistent throughout and remain unaffected by COVID?
Response: We thank you for the reviewer’s valuable comments.
Recent association studies explored the possible role of vitamin D deficiency. The low levels of 25(OH)D could be a consequence of COVID-19 (PMID: 32705585). Vitamin D supplements at therapeutic doses might not be dangerous to COVID-19 patients; it might even mitigate the exacerbation or severity of the disease. Considering the emergency of the current situation, sufficient evidence to support this claim has yet to be collected. The recommended intake for individuals at higher risk of COVID-19 is 10,000 IU/day for 1 to 2 weeks and subsequently 5000 IU daily, with target 25(OH)D concentrations of over 40 to 60 ng/mL [PMID: 32252338]. However, further studies are needed to better explore possible associations between vitamin D deficiency and COVID-19 morbidity and lethality, and assess if compensating such deficiency could avoid or mitigate the worst manifestations of COVID-19 (PMID: 32705585). Well-conducted randomized controlled trials of different forms of vitamin D supplementation in patients with COVID-19 remain an urgent need. (Line 622~634)
- Line 462-463: How does Vitamin D regulate renin expression?
Response: We thank you for the reviewer’s valuable comments.
It has been reported two decades ago that vitamin D markedly suppressed renin transcription by a VDR-mediated mechanism in cell cultures (PMID: 12122115). (Line 499) Thereafter, numerous clinical study also suggested that low plasma 25-hydroxyvitamin D levels may result in upregulation of the renin angiotensin system in otherwise healthy humans. However, clinical findings showed no benefit from 8 weeks of correcting vitamin D deficiency on RAS activity or BP (PMID: 20351344).
- Line 483-484: What does COVID have to do with mammary cell tumor invasion?
Response: We thank you for the reviewer’s valuable comments.
We delete the sentence for avoiding confusing the readers.
- Line 489-491: Why not give amiloride instead of Vitamin D? Why even mention it?
Response: We thank you for the reviewer’s valuable comments.
With respect to the potential risk of hyperkalaemia in AKI, amiloride is not appropriate instead of vitamin D. Amoloride and vitamin D can inhibit uPAR expression by an off-target mechanism. If vitamin D can drive uPAR suppression through the vitamin D receptor deserves further evaluation. (Line 526~529)
- Line 500-512: This is the first mention of autophagy. Why wasn’t it mentioned in a previous section? And what does Arf6-Erk-Nox have to do with COVID disease progression?
Response: We thank you for the reviewer’s valuable comments.
This paragraph describes the deleterious effect of elevated Ag II on kidney injury. So is an adequate place to put this statement. Since COVID-19 patients always have an activation RAAS and high level of Ag II, therefore, it may promote ROS production and podocyte apoptosis through activating ADP-ribosylation factor 6 (Arf6)-Erk1/2-Nox4 signaling. (Line 540~541)
- Lines 524-544: This section is a list of things Vitamin D does to reduce kidney damage but it needs to focus on the types and pathways of damage that result from COVID. Which pathways are prominent in COVID-related kidney damage?
Response: We thank you for the reviewer’s critical suggestions.
In this review article, we focus on the effect of vitamin D on suPAR-related AKI in COVID-19. Since vitamin D has pleiotropic effects on human pathology (PMID: 34065735, PMID: 26656443 ). It is not easy to categorize vitamin D effects into different types and pathways. Furthermore, we can’t differentiate which pathway is the most important one. Through its’ anti-inflammation and suPAR suppression effect, vitamin D may provide excellent beneficial effects on protecting kidneys in patients with COID-19. We have made a summary of the potential mechanisms of different types and pathways of kidney damage by which vitamin D may protect against kidney injury in COVID-19-related AKI in figure 4. (Line 560~580, 588~596)
Section 5
- Section 5.1 is not useful as the entire paper has already pointed out AKI as the target for therapy during COVID infection. Suggest keeping the first paragraph and deleting the entire second paragraph.
Response: We thank you for the reviewer’s valuable comments. We have deleted the second paragraph of 5.1.
- Section 5.2 is not useful as it provides no guidelines, testing, dosages, or other useful recommendations. It should be rewritten with currently approved clinical guidelines or at least suggested guidelines for when, how, what, and how much to administer to patients and if preexisting kidney disease or hypertensive patients need more.
Response: We thank you for the reviewer’s valuable comments. We add some more description about the possible dosage for vitamin D.
Recent association studies explored the possible role of vitamin D deficiency. The low levels of 25(OH)D could be a consequence of COVID-19 (PMID: 32705585). Vitamin D supplements at therapeutic doses might not be dangerous to COVID-19 patients; they might even mitigate the exacerbation or severity of the disease. Considering the emergency of the current situation, sufficient evidence to support this claim has yet to be proved. The recommended intake for individuals at higher risk of COVID-19 is 10,000 IU/day for 1 to 2 weeks and subsequently 5000 IU daily, with target 25(OH)D concentrations of over 40 to 60 ng/mL (PMID: 32252338). However, further studies are needed to better explore possible associations between vitamin D deficiency and COVID-19 morbidity and lethality and assess if compensating for such deficiency could avoid or mitigate the worst manifestations of COVID-19 (PMID: 32705585). Well-conducted randomized controlled trials of different forms and dosages of vitamin D supplementation in patients with COVID-19 remain an urgent need. (Line 622~634)

Round 2
Reviewer 1 Report
Thank you for the revision. However, little change has been made and the same fundamental flaws remain. This is not surprising as more research is needed. There is insufficient evidence to support the assertions made by the authors. Any suggestion of a potential role for vitamin D in COVID-related AKI and the linking of that with suPAR is far too premature.
The proposed links between vitamin D and suPAR, and the consequent effect on kidney function in COVID-19 is highly speculative. The authors have presented very little convincing evidence of the links. Most of the scientifically robust information is confirming that (i) suPAR could serve as a marker of clinical severity of COVID-19, which has been well-established and (ii) vitamin D deficiency can potentially worsen prognosis in COVID-19.
However, it is not very clear how the authors have combined these facts to account for AKI in patients with COVID-19 and suggest a therapeutic role for vitamin D in preventing AKI in COVID-19.
Much more scientific evidence for the implications is needed. As it stands, the manuscript might be appropriate for a journal focused on medical hypotheses.
Author Response
Response to Reviewer 1
Questions: Thank you for the revision. However, little change has been made and the same fundamental flaws remain. This is not surprising as more research is needed. There is insufficient evidence to support the assertions made by the authors. Any suggestion of a potential role for vitamin D in COVID-related AKI and the linking of that with suPAR is far too premature.
Response: Thank you very much for your efforts and time. We have all learned a lot from your criticism. As the reviewers state, any suggestion regarding the potential role of vitamin D in COVID-related AKI and linking it to suPAR is premature. At present, there is a lack of definitive data to confirm our speculation. There is also a lack of firm data on this interesting topic. For this reason, we conducted the narrative presumptive review based on the current lack of evidence to explore the possible links between suPAR, AKI, vitamin D, and COVID-19. I think this topic will be of interest to wider readers and will evoke the design of basic and clinical studies to further test this possible speculation. From our review information, we found that high levels of suPAR may bind to circulating uPA, which may lead to a hypercoagulable state, and that high levels of suPAR may activate intrarenal integrin signaling as in other diseases.
Questions: The proposed links between vitamin D and suPAR, and the consequent effect on kidney function in COVID-19 is highly speculative. The authors have presented very little convincing evidence of the links. Most of the scientifically robust information is confirming that (i) suPAR could serve as a marker of clinical severity of COVID-19, which has been well-established and (ii) vitamin D deficiency can potentially worsen prognosis in COVID-19.
Response: Thank you so much for your invaluable suggestions. I fully agree with the reviewer on most scientifically sound information that (i) suPAR can be a well-established marker of COVID-19 clinical severity, and (ii) vitamin D deficiency may worsen COVID-19 19 prognosis. In this perspective paper, we wish to organize these two scientific proofs into new knowledge. However, this assumption needs and merits further assessment. By the way, right now we're undertaking a clinical study to get the possible results.
Questions: However, it is not very clear how the authors have combined these facts to account for AKI in patients with COVID-19 and suggest a therapeutic role for vitamin D in preventing AKI in COVID-19.
Response: Thank you so much for your important feedback. There are many direct and indirect mechanisms and pathways that may be involved in COVID-19-related AKI. This hypothetical article discusses the suPRA-mediated AKI in the context of COVID-19 and the potential beneficial role of vitamin D. At this time, it is premature to conclude this presumption. Therefore, we changed the topic to "Potential Beneficial Effects of Vitamin D on suPAR-mediated IAR in COVID-19".
Questions: Much more scientific evidence for the implications is needed. As it stands, the manuscript might be appropriate for a journal focused on medical hypotheses.
Response: Many thanks for your valuable reviews. We agree with the need for additional scientific evidence to demonstrate its effects. Therefore, we offer narrative review articles to awaken interest from wider readers who can design different studies to shed light on this subject. Furthermore, the category of IJMS journals is multidisciplinary, and this quasi-medical hypothesis manuscript is suitable for the journal.

Reviewer 2 Report
The authors are to be commended for making all recommended changes.
Author Response
Response to Reviewer 2
Moderate English changes required
Response: Thank you very much for your efforts and time. We have corrected the English editing of the entire manuscript at your request.
The authors are to be commended for making all recommended changes.
Response: Many thanks to the reviewers for agreeing with our response.
